# Assessment of GGE, AMMI, Regression, and Its Deviation Model to Identify Stable Rice Hybrids in Bangladesh

**DOI:** 10.3390/plants11182336

**Published:** 2022-09-07

**Authors:** M Jamil Hasan, M Umma Kulsum, Umakanta Sarker, M Quamrul Islam Matin, Nazmul Hoque Shahin, M Shahjahan Kabir, Sezai Ercisli, Romina Alina Marc

**Affiliations:** 1Hybrid Rice Division, Bangladesh Rice Research Institute, Gazipur 1701, Bangladesh; 2Department of Genetics and Plant Breeding, Faculty of Agriculture, Bangabandhu Sheikh Mujibur Rahman Agricultural University, Gazipur 1706, Bangladesh; 3DG Office, Bangladesh Agricultural Research Institute, Gazipur 1701, Bangladesh; 4Department of Agriculture, Bangabandhu Sheikh Mujibur Rahman Science and Technology University, Gopalganj 8100, Bangladesh; 5DG Office, Bangladesh Rice Research Institute, Gazipur 1701, Bangladesh; 6Department of Horticulture, Faculty of Agriculture, Ataturk University, Erzurum 25240, Turkey; 7Food Engineering Department, Faculty of Food Science and Technology, University of Agricultural Sciences and Veterinary Medicine, 400372 Cluj-Napoca, Romania

**Keywords:** boxplot, histogram, skewness, kurtosis, hybrid combination

## Abstract

Twenty-six rice hybrids were evaluated at three locations. Data were normally distributed after running the Shapiro–Wilk test. Plant height and effective tillers/hills showed leptokurtic distribution, indicating these traits were controlled by fewer genes, whereas the rest of the attributes had platykurtic distribution, indicating these traits were controlled by many genes. Most of the traits were significant for variety, locations, and variety × locations. For yield stability, the data were analyzed using additive main effect and multiplicative interaction (AMMI), genotype and genotype–environment interaction (GGE), and Eberhart and Russell’s model. Among 26 hybrids, BRRI99A × BRRI38R and BRRI hybrid dhan5 exhibited high yields at three locations. BRRI99A × BRRI45R, BRRI99A × BRRI31R, IR79156A × BRRI38R, and BRRI hybrid dhan3 were selected for mega-environments: Gazipur and Ishwardi. Among the tested locations, Gazipur (E2) and Ishwardi (E3) were identified as mega-environments for the hybrid combinations, including BRRI99A × BRRI36R, BRRI99A × BRRI49R, IR79156A × BRRI31R, IR79156A × BRRI38R, BRRI hybrid dhan5, BRRI99A × BRRI38R, BRRI99A × BRRI45R, and BRRI99A × BRRI31R based on their average action and fixity. Gazipur and Ishwardi were the best environments because their discriminative and representative ability was remarkable. The hybrid assessment, as well as area selection for hybrid rice breeding in Bangladesh, were revealed in this study. The hybrid BRRI99A × BRRI38R, BRRI99A × BRRI36R, and IR79156A × Rline7 belonged to medium-to-long slender grain types. Nowadays, the citizens of Bangladesh prefer fine-grain rice. Therefore, these fine-grain hybrids can be cultivated as preferable commercial varieties at three locations, such as Barisal, Gazipur, and Ishwardi in Bangladesh. The stable hybrids identified in the current study can be recommended for cultivation throughout the whole country without compromising the loss of grain yield of rice.

## 1. Introduction

Rice in Bangladesh is a major food crop. The production of the present variety has become stable. To feed the burgeoning population in this country, increasing rice yield vertically is the chief mission for future demand. Hybrid rice is one of the choices for increasing yield per unit area in this situation because it yields 15–20% more than existing varieties [1].

Descriptive statistical measures viz., mean, median, range, quartiles, standard deviation, standard error, skewness, and kurtosis can provide a lot of information about any variable of interest. Yield is a complex qualitative character with strong environmental interaction. Selection of variety based on a single environmental performance is not effective [2]; it varies from one environment to another. It needs a selection of variety in the multi-environmental situation [3]. Stability means the firmness of variety in varying environments, whereas adaptability refers to the better survivability of a genotype in the specific environment. The best specific environment was preferred for massive area cultivation in that case, besides higher production yield stability, which is also fascinating. Stable genotypes are essential for an extensive array of cultivation over locations, inputs, and times [4,5,6,7,8]. To detect such genotypes, G × E interaction is a serious headache for a breeder because it confines the choice of the best variety by changing their relative productiveness in various environments [9,10,11,12,13,14].

The best adapted genotype to the specific environment and a graphical representation of the relationship among the genotype, environment, and genotype–environment interaction (GEI) were interpreted using biplots. Recently additive main effects and multiplicative interaction (AMMI) biplot and genotype and genotype-by-environment (GGE) biplots were broadly used for biplot graphs. Yan et al. [15] mentioned that GGE biplots are the standard model for genotype and environment evaluation. Picking out the mega-environment, studying the ranking of genotypes, and noticing the discriminative and determinative power of the studied environments were represented using the GGE biplot mentioned by some researchers [16]. It may important to specify a favorable area for hybrid rice cultivation in Bangladesh. As a result, seeking the best hybrid rice variety with more yield and stability was the goal of this experiment.

## 2. Results and Discussion

ANOVA displayed a significant variation among the traits. An extensive length of differences was also stated in yield and their associated characteristics of rice [17,18,19,20,21,22,23,24,25,26,27,28,29,30,31,32,33,34], corn [35,36,37], lady’s finger [38,39,40], coconut [41,42], broccoli [43], and biochemical traits of amaranth [44,45,46,47,48,49,50,51], and other crops [52,53,54] which is corroborated by our present findings.

Experiments were carried out in three different locations: Gazipur, Barisal, and Ishwardi. Data indicate that the sample of populations collected was normally distributed. That is why the probable strategies require checking the normality acceptance. The Shapiro–Wilk test was used for the normality test of the data, and the results were normal. One of the most-often-used tests for normalcy is the Shapiro–Wilk test.

Descriptive statistics evaluate a large amount of data on all variables of interest. Days to maturity ranged from 144 (Teea) to 156 (Heera-2) with an average of 150 (Figure 1), plant height ranged from 107 cm (BRRI99A × BRRI36R) to 118 cm (H-386) with an average of 110 cm, effective tillers/hills ranged from 9 (BRRI35A × BRRI49R) to 13 (TejGold) with an average of 12, spikelet fertility (%) ranged from 83% (IR79156A × BRRI49R) to 89% (BRRI99A × BRRI36R) with an average of 86%, with spikelet fertility having a greater influence on yield. Panicle length ranged from 26 cm (IR79156A × BRRI49R) to 30 cm (BRRI99A × BRRI38R) with an average of 28 cm, and flag leaf length ranged from 27 cm (BRRI35A × BRRI49R) to 32 cm (BRRI99A × BRRI31R) with an average of 30 cm, with flag leaf length having a major influence on grain filling. Thousand-grain weight ranged from 24 g (IR79156A × BRRI49R) to 28 g (BRRI99A × BRRI31R) with an average of 27 g, and yield ranged from 8 t/ha (IR79156A × BRRI49R) to 10 t/ha (BRRI99A × BRRI38R) with an average of 9 t/ha.

### 2.1. Box Plot Sketch Depends on Descriptive Statistics

Descriptive information-based box plots for every character are given in Figure 1, which visualizes the distribution of data. With this plot, we can get a complete idea of the symmetry of the data. In addition, it provides the concept of the scatteredness of data. This plot was used to realize the normality pattern of the data. Five types of summaries are shown in the box plot, where the first quartile, second quartile or median, third quartile, and minimum and maximum values are exposed. Here we tried to plot our data in a box whose midpoint is the sample median, the top of the box is the third quartile (Q3), and the bottom of the box is the first quartile (Q1). The upper whisker extends to this adjacent value—the highest data value within the upper limit = Q3 + 1.5 IQR, where the interquartile range IQR is defined as IQR = Q3 − Q1. Similarly, the lower whisker extends to this adjacent value—the lowest value within the lower limit = Q1 − 1.5 IQR. When the large or small data is plotted behind the whiskers they are treated as outliers [55].

Among the 26 hybrid combinations, 75% of individuals had equal to or less than 151 (Q3) days to maturity, whereas 25% of genotypes had equal to or less than 147 (Q1) days to maturity. Similarly, 75% of hybrid combinations recorded equal to or less than 111 cm, whereas 25% of hybrid combinations have equal to or less than 109 cm plant height. In this way, 75% of effective tillers/hills were within 12 (3rd quartile) and 25% were within 11 (1st quartile). However, 75% of spikelet fertility (%) showed equal to or less than 87% (3rd quartile) and 25% of hybrid combinations had 85% equal to or less than 25% (1st quartile). For panicle length, 75% of combinations had equal to or less than 29 cm (3rd quartile) and 25% of combinations had 27 cm (1st quartile). In the case of flag leaf-length, 75% of hybrid combinations had equal to or less than 31 cm (3rd quartile) and 25% of hybrid combinations had equal to or less than 28 cm (1st quartile). In this way, for thousand-grain weight 75% of combinations showed equal to or less than 28 g (3rd quartile) and 25% of the combination were equal to or less than 26 g (1st quartile). Yield showed that 75% hybrid combinations were equal to or less than 9 t/ha and 25% hybrid combinations were equal to or less than 8 t/ha. The middle value of each characteristic was nearly identical or closer to the particular mean value, pointing out the right ordination of genotypes for comparing traits [56].

Plant height and effective tillers/hills both had outliers in their data. Plant height was skewed to the right, whereas effective tillers/hills was biased to the left. Imon and Das [57] also showed a similar result in the case of an outlier.

### 2.2. Frequency Distribution-Based Histogram of Some Hybrid Combinations

A histogram depicts the easiest and simplest visual distribution of data (Table 1 and Figure 2). The traits’ values were plotted against their frequency to determine whether the distribution was bell-shaped or not. It also includes information on the data’s insight gap and outliers. The distribution frequency of skewness and kurtosis gives information about the nature of gene actions and the number of genes involved in the traits [58]. As a result of linkage and variable crossover, the skewness and kurtosis values for each trait and population can change and result in different genome recovery of the parents. A large number of genes controlled a trait, with the majority of them demonstrating dominant and dominant-based duplicate epistasis (additive × additive gene action) for negatively skewed platykurtic distribution. Although the platykurtic distribution was favorably skewed, it indicates that this trait was driven by a large number of genes, with the majority of them demonstrating dominant and dominant-based complementary gene action. Positive kurtosis (leptokurtic) suggests few genes regulate the trait (mono or oligogenic inheritance, i.e., less environmental influence on the trait), whereas normal distribution at zero (mesokurtic) and negative kurtosis (platykurtic) suggests many genes regulate the trait (polygenic inheritance, i.e., large environmental influence on the trait) [59,60]. From this study, days to maturity had positive skewness (0.44) and negative kurtosis (−0.49). Plant height exhibited skewness and kurtosis coefficients of 1.71 and 3.24, respectively, which displayed rightly skewed and leptokurtic distribution. Effective tillers/hills exhibited negatively skewed (−1.69) and leptokurtic distribution (kurtosis = 4.47). Spikelet fertility % showed positive skewness (0.09) and kurtosis (0.18) values, which were close to a mesokurtic configuration. Panicle length had positive skewness (0.25) and negative kurtosis (−0.17). These findings on the skewness and kurtosis of plant height are supported by Raghavendra and Hittalmani [61]. Flag leaf length had negative skewness (−0.19) and kurtosis (−1.19). The thousand-grain weight and grain yield also showed negative skewness (−0.41 and −0.08) and kurtosis (−0.18 and −0.21) values, which indicate duplicate gene interaction controlled by a larger number of genes (polygenic inheritance) with large environmental influence. The results of the seed yield and 100-seed weight of Dinesh et al. [62] were corroborated by our thousand-grain weight and grain yield. The positive coefficient of kurtosis for plant height and effective tillers/hills indicates the presence of a small number of dominant genes with monogenic or oligogenic inheritance, i.e., less environmental influence on these traits [63].

### 2.3. Analysis of Variance

Within variety and location, yield and yield-related traits showed a broad range of changes. A combined analysis of variance of data from three locations displayed significant variety and locations for most of the traits (Table 2). In the case of location, effective tillers/hill was non-significant. Location × variety interaction was significant for panicle length, days to maturity, grain yield, and thousand-grain weight.

Every location had three replications for analyzing and comparing the data with the high-yield test. Among the hybrid combinations, the highest DTM was recorded for BRRI35A × BRRI49R in Gazipur and Ishwardi, when it was compared with check variety Heera-2. Heera-2 showed the highest DTM in all three locations. The combination BRRI99A × BRRI36R had the lowest growth duration, and it was compared with check variety Teea. Teea showed the lowest growth duration in the three locations. A shorter growth duration was our desired characteristic. This result was similar to Hasan et al. [64]. The highest PH was recorded in check variety H386 for all locations, and dwarf plant type was recorded in hybrid combination BRRI99A × BRRI36R in all locations. The dwarf variety is considered to display lodging resistance [65]. The variety H-386 gave the highest effective tillers/hills at the Barisal location, IR79156A × BRRI45R at Gazipur, and Tejgold at Ishwardi. Hybrid combination BRRI99A × BRRI36R showed the highest SF% at Gazipur and Ishwardi compared with check variety BRRI hybrid dhan5 but not in Barisal. The highest PL was observed in BRRI99A × BRRI38R in all locations, followed by a check variety BRRI hybrid dhan5. Therefore, in comparison with BRRI hybrid dhan5, the hybrid combination BRRI99A × BRRI38R was selected for all locations. In the case of flag leaf length (FLL), the highest FLL was recorded in BRRI99A × BRRI36R followed by H-2264 at Barisal, but in Gazipur the highest was in H-2264, followed by IR79156A × BRRI31R. In Ishwardi the highest FLL was BRRI99A × BRRI31R. The highest thousand-grain weight (TGW) was recorded in BRRI99A × BRRI36R compared with check variety BRRI hybrid dhan5 in all locations. The hybrid combination BRRI99A × CHA15R was identified as the highest grain yield followed by BRRI hybrid dhan5 at Barisal and Ishwardi, but in Gazipur the highest grain yield was recorded in BRRI99A × BRRI36R followed by BRRI hybrid dhan5. the grain yields and related traits for all three locations were then put down to combine the analysis with AMMI and GGE biplot modes. Specific genotypic adaptation and general genotypic adaptation to different environments were identified by stability analysis using PB tools and R software after analysis of variance.

### 2.4. General Genotypic Adaptation

AMMI and GGE biplots were used for the interpretation of general genotypic adaptability (Figure 3). The biplot was used to notice the contribution of different genotypes in an approved environment. The relative arrangement of different genotypes on biplots based on its projection on the O-axis in the AMMI biplot and the GGE biplot was used to diagnose G × E interaction effects on each trait contributing to grain yield. The main effect of genotypes and environments and the first multiplication axis (PC1) were plotted in the AMMI biplot graph, which interpreted the AMMI model. Higher principal component axis (PC1) scores, whether negative or positive, indicate specific adaptation of a genotype to specific environments. The closer the PC1 scores are to zero, the more stable the genotype is among the environments that are studied. Among the hybrid combinations BRRI48A × BRRI38R, BRRI99A × BRRI38R, IR79156A × BRRI38R, BRRI99A × BRRI45R, IR79156A × BRRI45R, BRRI99A × BRRI31R, BRRI hybrid dhan3, and BRRI hybrid dhan5 exhibited high grain yield with high main (additive) effects showing a positive PCI score. These hybrid combinations also showed stable performance. Environmental interaction was low and grain yield interaction was high among hybrid combinations BRRI35A × BRRI36R, Teea, H-2264, JhonokRaj, and Heera-2. Therefore, the IR79156A × R line7 showed high interaction with the environment of Gazipur. Hybrid combination BRRI99A × BRRI36R, BRRI99A × BRRI49R, and IR79156A × BRRI31R was high yielding but unstable due to their low PCI score. Some combinations of BRRI35A × BRRI45R, BRRI35A × BRRI52R, TejGold, Gold, and SL8H were low yielding but the PCI score was high, and BRRI35A × BRRI49R, IR79156A × BRRI49R, and BRRI35A × BRRI37R where low yielding and the PCI score was lower. These results were identical to those of Matin et al. [66]. Based on AMMI analysis, IR79156A × BRRI49R, BRRI48A × BRRI38R, and IR79156A × BRRI38R were the most stable genotypes for days to maturity with medium growth duration; BRRI99A × BRRI49R and BRRI99A × BRRI31R for effective tillers/hills; BRRI99A × BRRI45R and Gold for flag leaf length; IR79156A × BRRI49R, IR79156A × BRRI38R, and IR79156A × BRRI45R for plant height; BRRI99A × BRRI38R and BRRI hybrid dhan5 for panicle length; IR79156A × BRRI38R, BRRI99A × BRRI49R, and BRRI99A × BRRI45R for spikelet fertility (%); and BRRI99A × BRRI36R, Heera-2, BRRI hybrid dhan3, and Gold for thousand-grain weight. The stability of the hybrid combinations all over the environments was also demonstrated by GGE biplot (Figure 3).

### 2.5. Evaluation of Environment through the Which-Won-Where Pattern

The polygon idea of the GGE biplot visually identified genotypes by observing the genotypes’ “point angle” in every artificial area. Mega-environments along the specific location identification need the idea of the “which-won-where” model in multilocation yield trials. Figure 4 represents the 26 hybrid combinations that were evaluated in three locations. The polygon scene of the GGE biplot illustrated 96.23% of the variation of GGE for grain yield. This percentage explained that the first two principal components of the biplot adequately exhibited the GGE model for grain yield. In this figure (Figure 4) ray1 perpendicularly intersected the polygon on the side joining BRRI99A × BRRI36R and BRRI hybrid dhan5, ray2 bisected BRRI99A × BRRI36R and BRRI35A × BRRI37R, ray3 intersected the side between BRRI35A × BRRI37R and IR79156A × BRRI49R, and so on. The nine rays originated from the origin of the biplot. When splitting the polygon into sections, every section has its combination at the vertex. Akter et al. [67] also found three genotypes in the first section. The vertex hybrid combinations were BRRI99A × BRRI36R, BRRI hybrid dhan5, BRRI99A × BRRI38R, IR79156A × BRRI45R, H-386, SL8H, BRRI35A × BRRI49R, IR79156A × BRRI49R, and BRRI35A × BRRI37R with the largest distance from the origin. These hybrid combinations were the best or poorest hybrids in some or all environments because they were the farthest from the origin of the biplot [12], meaning they were more responsive to environmental change and are considered especially adapted hybrid combinations. If all environments fall into a single section and genotypes across the environments indicated, all genotypes will perform better. On the other hand, if the environment falls into different sections, the genotypes on the vertex of each section perform better in all the environments of that section. Crossover patterns or genotypes per environment interactions can be described in this way. This interaction displayed by the selected environment can be divided into several mega-environments. In Figure 4, E2 (Gazipur) and E3 (Ishwardi) comprised a mega-environment, with the hybrid combination BRRI99A × BRRI36R, BRRI99A × BRRI38R, and BRRI hybrid dhan5 as the wining cultivar in this mega-environment. Oladosu et al. [68] identified two mega-environments for grain yield in their tested environment. A mega-environment can be defined as a group of environments that always take part in the best set of genotypes across the year [69]. Test locations were grouped based on the polygon (which-won-where) idea of the GGE biplot, and BRRI99A × BRRI36R, BRRI99A × BRRI38R, and BRRI hybrid dhan5 were recommended in the mega-environment (Gazipur and Ishwardi) and the vertex hybrid combination IR79156A × BRRI45R, H-386, and SL8H for Barisal. Visualization of the which-won-where pattern explained the existence of several mega-environments of the hybrid rice-growing area in Bangladesh. This particular adjustment indicated a tall cruel efficiency of a genotype in a chosen environment [70]. The hybrids JhonokRaj and Heera-2 showed more stability across all environments when they were near the origin. The present findings are in harmony with Matin et al. [71]. Genotypes with high PCI values showed considerable interactions all over the environments, but genotypes with PCI scores near zero showed little interaction in the overall environments. Balakrishnan et al. [72] also found that genotypes G2, G6, and G12 showed fewer environmental interactions.

### 2.6. Genotype Discrimination by GGE Biplot in Mean versus Stability Views

Genotype positions were identified in the average-environment axis (AEA) or average-environment coordination (AEC) from the GGE biplot based on their mean and stabilities (Figure 5). Two lines were shown in this graph, the AEC abscissa (vertical line) and the AEC ordinate (horizontal line). AXIS1 and AXIS2 (the environmental variables) are plotted on two separate axes; the biplot origin is a hypothetical average environment (Figure 5). The arrowhead direction on the AEA abscissa points towards a higher mean value for grain-yield performance. This graph shows that the BRRI hybrid dhan5, BRRI99A × BRRI38R, BRRI99A × BRRI45R, BRRI99A × BRRI31R, and BRRI99A × BRRI36R combinations were the highest yielders. The second line in this graph measures stability. This line in the AEC ordinate opposite the AEC abscissa overcoming the biplot center implies the genotypes’ stability. The largest vector from AEC means high variability or instability in either direction. The shortest vector from AEC means low variability or a more stable hybrid combination among the tested environments and vice versa. Hence hybrid BRRI hybrid dhan5 was the most stable, whereas BRRI35A × BRRI37R was the most unstable among the hybrids. High mean and highly stable hybrid combinations were ideal for selection. This biplot showed that the hybrid combinations BRRI hybrid dhan5, BRRI99A × BRRI38R, BRRI99A × BRRI45R, BRRI99A × BRRI31R, and IR79156A × BRRI38R could be stated as ideal due to the comparatively shorter distance from AEC and closeness to the small circle. Hashim et al. [73] stated that G1 had the highest grain yield but low stability, whereas G15 was highly stable and had a low grain yield. Grain yield performance can primarily be measured in two mean ideas and stability; in this way a breeder can use information by choosing a genotype with its adaptation to specific environments [15].

### 2.7. Discriminative vs. Representative

The selection of superior genotypes and the identification of a suitable test environment are both critical breeding strategies. The two criteria of an ideal environment are its representativeness and discriminative power in all environments (Figure 6). The ability of an environment or place to define genotypes is known as discriminative power, whereas representativeness refers to the ability of a tested environment to reflect the other tested environment.

The vector length of the tested environment gives a measurement of its discriminating power, which specifies the genotype of an environment. According to this biplot, the short vector environment of Gazipur (E2) and Ishwardi (E3) for grain yield can be determined as a self-reliant research location and also behave as a unique location. Meanwhile, the long vector location is Barisal (E1), which was more influential in discriminating among the hybrid combinations. Environment E2 contains a short vector, and tiny angles with AEC abscissa are an ideal environment for a superior hybrid combination. Therefore, in the study E2 was selected for the hybrid combination BRRI99A × BRRI38R.

### 2.8. Ranking Genotypes

Figure 7 presents the ranking of a hybrid combination relative to an ideal hybrid combination with a concentric circle [68]. Hence, the ideal genotype was conceptually located at the center, in which genotypes were diagrammed to demonstrate the correlation between other genotypes and distance from the ideal genotype. In this figure, BRRI hybrid dhan5 falls within the center of the concentric circle, and the hybrid combinations BRRI99A × BRRI45R, BRRI99A × BRRI38R, BRRI99A × BRRI31R, and IR79156A × BRRI38R are at a short distance from the concentric circle for grain yield.

### 2.9. Grain Type

Classification of grain shape was measured based on standard evaluation systems (SES) of IRRI [74] for all the hybrid combinations given in Table 3. In a total of 26 hybrid combinations, 7 combinations showed a medium slender, 7 had a long slender, 11 combinations had a medium bold, and 1 had a long bold-type grain shape. Future breeding programs need to focus on stress-tolerant, nutrient-contained, and high-yielding hybrid combinations.

### 2.10. Stability Analysis for Different Characters of 26 Promising Hybrid Rice Grains

Eberhart and Russel [75] emphasized the need for both linear (bi) and non-linear (S^2^di) components of genotype x environment interactions in judging the phenotypic stability of a genotype. In this model, regression coefficient (bi) is considered a parameter of response, and deviation from regression (S^2^di) is the parameter of stability. The relatively lower value of bi (around 1) means less responsiveness to environmental change and therefore, more adaptiveness. If bi is negative, the genotype may be grown only in a poor environment. If S^2^di is significantly different from zero, it invalidates the linear prediction. If S^2^di is non-significant, the performances of a genotype for a given environment may be predicted. Therefore, a genotype whose performance for a given environment can be predicted if S^2^di ~0 and it is a stable genotype. The Ij directly reflects the poor or rich environment in terms of negative and positive Ij, respectively.

For days to maturity the hybrid combinations BRRI99A × BRRI36R, BRRI99A × BRRI49R, BRRI35A × BRRI37R, BRRI99A × BRRI31R, IR79156A × BRRI31R, BRRI hybrid dhan3, TejGold, Gold, and Teea showed negative Pi, non-significant bi, and S^2^di values indicating the hybrid combinations were stable at all locations with medium growth duration (Table 4). The hybrid combination BRRI35A × BRRI36R displayed a negative Pi, significant bi, and an S^2^di value of zero, which indicates stability in a specific location. The hybrid combinations BRRI35A × BRRI49R, IR79156A × BRRI49R, BRRI48A × BRRI38R, BRRI99A × BRRI38R, IR79156A × BRRI38R, BRRI35A × BRRI45R, BRRI99A × BRRI45R, H-2264, H-386, JhonokRaj, and Heera-2 had positive Pi and non-significant bi and S^2^di values, suggesting stable hybrid combinations with higher growth duration. The hybrid combinations IR79156A × R line7 and IR79156A × BRRI45R showed positive Pi, negative and non-significant bi, and positive and non-significant S^2^di values, indicating that they were grown only in the unfavorable location of Gazipur. BRRI35A × BRRI52R and BRRI hybrid dhan5 also showed negative Pi, negative and non-significant bi, and positive and non-significant S^2^di values, and as a result they were reactive to the unfavorable location of Gazipur with short growth duration. The environmental index (Ij) directly reflects the rich or poor environment in terms of negative and positive Ij, respectively, for days to maturity (Table 4).

Hybrids BRRI35A × BRRI36R, BRRI99A × BRRI36R, BRRI35A × BRRI49R, BRRI35A × BRRI37R, BRRI99A × BRRI38R, BRRI99A × BRRI45R, BRRI35A × BRRI52R, BRRI99A × BRRI31R, BRRI hybrid dhan3, BRRI hybrid dhan5, TejGold, JhonokRaj, Gold, Teea, and SL8H showed negative Pi for plant height and non-significant bi and S^2^di values, which indicates the stability of hybrid combinations with shorter plant stature (Table 5). A shorter variety is required to maintain lodging resistance. The combinations BRRI99A × BRRI49R had negative Pi, significant bi, and S^2^di~0, which reveals that this hybrid was suitable for the specific location of Gazipur. BRRI48A × BRRI38R and Heera-2 obtained negative Pi, negative and non-significant bi, and positive and non-significant S^2^di, indicating that the hybrids were stable in all locations with a semi-dwarf plant stature. IR79156A × BRRI45R had a positive Pi, negative and non-significant bi, and positive and non-significant S^2^di values, alluding to this hybrid being responsive to the location of Gazipur with a higher plant stature (Table 5).

For the effective tillers/hills, the positive Pi represents the higher number of tillers/hills and the negative Pi represents the lower number of tillers/hills among the hybrid combinations. Again, positive and negative Ij reflects the rich or favorable and poor or unfavorable environments for this characteristic, respectively (Table 6). The 15 hybrid combinations BRRI99A × BRRI49R, BRRI48A × BRRI38R, BRRI99A × BRRI45R, BRRI35A × BRRI52R, BRRI99A × BRRI31R, H-2264, H-386, BRRI hybrid dhan3, BRRI hybrid dhan5, TejGold, JhonokRaj, Heera-2, Gold, and SL8H showed a positive Pi and non-significant bi and S^2^di values, indicating stable effective tillers/hills for all the locations. IR79156A × BRRI45R showed a positive Pi, negative and non-significant bi, and positive and non-significant S^2^di values, which show that this hybrid was reactive to the locations of Gazipur and Ishwardi (Table 6).

A positive Pi for spikelet fertility percentage indicated a crucial part of increasing yield and a negative Pi showed a decrease in the yield performance. The Ij directly reflects the poor or rich environment in terms of negative and positive Ij, respectively. Thus, the location of Ishwardi was favorable for spikelet fertility percentage, and Gazipur and Barisal were unfavorable for spikelet fertility percentage. Considering the mean, bi, and S^2^di, it is evident that all the hybrid combinations exhibited different responses of adaptability under different environmental conditions. BRRI99A × BRRI36R, BRRI99A × BRRI49R, BRRI35A × BRRI37R, BRRI99A × BRRI38R, IR79156A × BRRI38R, BRRI99A × BRRI45R, BRRI99A × BRRI31R, BRRI hybrid dhan5, TejGold, and JhonokRaj had positive Pi and non-significant bi and S^2^di values, indicating that these hybrid combinations were stable overall in the locations of Barisal, Gazipur, and Ishwardi (Table 7). Low mean and negative Pi with non-significant bi and non-significant S^2^di were acquainted with the hybrid combinations BRRI35A × BRRI36R, BRRI35A × BRRI49R, IR79156A × BRRI49R, IR79156A × R line7, BRRI35A × BRRI45R, IR79156A × BRRI45R, BRRI35A × BRRI52R, IR79156A × BRRI31R, H-2264, H-386, Heera-2, Gold, Teea, and SL8H. These hybrids were stable but cannot be considered for those locations due to their lower spikelet fertility percentage. BRRI48A × BRRI38R and BRRI hybrid dhan3 showed positive Pi, negative and non-significant bi, and positive and non-significant S^2^di values, meaning they were responsive to the locations of Barisal and Gazipur (Table 7).

A positive Pi and higher mean for panicle length had an evidentiary contribution to increasing yield and a negative Pi indicated decreasing yield. The mean, bi, and S^2^di values were taken and the genotypic response and adaptability under different environmental conditions were enumerated (Table 8). The hybrid combinations BRRI99A × BRRI36R, BRRI99A × BRRI49R, BRRI99A × BRRI38R, BRRI99A × BRRI45R, IR79156A × BRRI45R, BRRI99A × BRRI31R, H-386, BRRI hybrid dhan3, BRRI hybrid dhan5, JhonokRaj, Heera-2, and Gold showed positive Pi and non-significant bi and S^2^di values, pointing out stable performance for panicle length in all the locations. Gazipur and Ishwardi were the favorable locations and Barisal was the unfavorable location for panicle length (Table 8).

High mean and positive Pi for flag leaf length covered more area and increased photosynthesis, which ultimately increased yield, but lower mean and negative Pi had lower yield performance. The mean, bi, and S^2^di were considered as the different responses of adaptability in different locations for the hybrid combinations. The hybrid combinations BRRI35A × BRRI36R, BRRI48A × BRRI38R, BRRI99A × BRRI38R, IR79156A × BRRI38R, BRRI99A × BRRI45R, IR79156A × BRRI31R, BRRI hybrid dhan5, and Gold had positive Pi and non-significant bi and S^2^di values, which conveyed the stability of these hybrids under all location for flag leaf length (Table 9). Gazipur and Ishwardi were the positive locations and Barisal was the negative location for flag leaf length. BRRI99A × BRRI36R, BRRI99A × BRRI49R, IR79156A × BRRI45R, BRRI99A × BRRI31R, and H-2264 showed positive Pi, negative and non-significant bi, and S^2^di near zero, which suggests that these hybrid combinations were highly responsive to the location of Barisal (Table 9).

Higher thousand-grain weight and positive Pi confirmed higher yield. The bi values for thousand-grain weight ranged from −6.510 to 9.297. These differences in bi values indicate that all hybrid combinations responded differently to different environments. High and considerably positive Pi, non-significant bi, and S^2^di were recorded in the hybrids BRRI99A × BRRI36R, BRRI35A × BRRI49R, BRRI99A × BRRI49R, BRRI99A × BRRI38R, IR79156A × BRRI38R, BRRI99A × BRRI31R, BRRI hybrid dhan3, JhonokRaj, Heera-2, and Gold, which signals that the hybrids were stable over all the locations (Table 10). The hybrid combination BRRI99A × BRRI45R displayed a positive Pi, significant bi, and S^2^di of zero (S^2^di ~0), alluding to the thousand-grain weight being highly responsive to the locations of Gazipur and Ishwardi. BRRI35A × BRRI36R, IR79156A × BRRI45R, H-2264, BRRI hybrid dhan5, and Teea showed positive Pi, negative and non-significant bi, and S^2^di near zero, giving a hint that these hybrids were reactive to the poor location of Barisal (Table 10).

A high mean and positive Pi for grain yield was one of our desired characteristics for measuring stability. The hybrid combinations BRRI99A × BRRI36R, BRRI99A × BRRI49R, IR79156A × BRRI38R, IR79156A × BRRI31R, BRRI hybrid dhan3, JhonokRaj, and Heera-2 showed positive Pi and non-significant bi and S^2^di values, notifying that they were stable over all locations. BRRI99A × BRRI45R had a higher yield and was highly responsive to Barisal location. Kulsum et. al. [65] observed similar findings in rice in their previous study. BRRI35A × BRRI36R, BRRI35A × BRRI49R, IR79156A × BRRI49R, BRRI35A × BRRI37R, IR79156A × R line7, BRRI35A × BRRI45R, BRRI35A × BRRI52R, H-2264, Teea, and SL8H gained low mean and negative Pi and non-significant bi and S^2^di values in these hybrids, showing that they were stable all the locations, but their stability was not considered because of lower yield performance (Table 11). Gazipur and Ishwardi were favorable locations and Barisal was the unfavorable location for grain yield. The hybrid combinations BRRI48A × BRRI38R, BRRI99A × BRRI38R, IR79156A × BRRI45R, BRRI99A × BRRI31R, H-386, and BRRI hybrid dhan5 contained positive Pi, negative and non-significant bi, and positive and non-significant S^2^di, revealing that these hybrids were responsive to the location of Barisal (Table 11).

From the above discussion, yield and most of the yield-contributing characteristics explained that the location of Barisal was poor, and that Gazipur and Ishwardi were favorable for hybrid rice production. The most stable hybrid combinations were BRRI99A × BRRI36R, BRRI99A × BRRI49R, IR79156A × BRRI38R, BRRI99A × BRRI45R, BRRI hybrid dhan3, BRRI hybrid dhan5, JhonokRaj, and Heera-2 across all locations.

## 3. Materials and Methods

### 3.1. Materials of the Study

This study was conducted at three various agro-ecological zones (AEZ) of Bangladesh, including Barisal, Gazipur, and Ishwardi, during Boro 2019–2020. Climatic conditions of these three zones are presented in Appendix A. Twenty-six hybrids were evaluated; among these, 16 were new hybrid combinations and 10 were release hybrids (Table 12).

### 3.2. Layout and Design

The layout of this experiment was a randomized complete block design (RCBD) with 3 replications in each environment. The age of the seedlings was 30 days for transplanting; one seedling was used on one hill at the line to line at 20 cm and the hill to hill at 15 cm. Every experimental plot was 30 square meters.

### 3.3. Intercultural Practices

Fertilizer nourishment practices and plant cultural measures were based on Adhunik Dhaner Chash BRRI [76] recommendations for each area. Two border lines were employed to minimize the border impact.

### 3.4. Data Collection

For measuring data on days to maturity, plant height, effective tiller/hill, spikelet fertility, panicle length, flag leaf length, thousand-grain weight, and grain yield the standard evaluation systems of IRRI [74] were followed, and grain yield was calculated using total plot yield and then converted to t/ha at 14% moisture content.

### 3.5. Statistical Analysis

Narrative statistics for all the character’s boxplots and histograms were prepared by using Excel v 2019. Statistics10 software was used for calculating the analysis of variance. PB tools (Version 1.4, http://bbi.irri.org/products) (accessed on 1 March 2022) and R (R Core Team [77]) were used for the combined analysis of variance in both genotypes and environments. The R studio package of graphical user interface (GUI) was used for GGE biplots, consisting of two concepts, the biplot concept [78] and the GGE concept [15]. A GGE biplot narrates the visual analysis of multi-environment trial (MET) data [79]. Mean and stability were measured by GGE biplots for setout G × E interaction and genotype ranking. Mega-environment rating (which-won-where pattern), genotype rating (mean versus stability), and studied environment ranking (discriminative versus representative) were visualized by a graph. The error means square was used to test the significance of all outcomes. The action of hybrids was turned over three locations using (1) the stability models additive main effects and multiplicative interaction (AMMI) [80], and (2) GGE biplot [81] and Eberhart and Russell’s model [75]. The stability and GEI patterns were interpreted and visualized using these models. Only GEI styles were overwhelmed by the AMMI model of the multiplicative component. On the other hand, both genotypic effect and GEI were overwhelmed by the GGE model of the multiplicative component.

## 4. Conclusions

The results of the Shapiro–Wilk test were normal, according to the study. Plant height and effective tiller/hill had leptokurtic distributions, whereas the rest of the traits had platykurtic distributions. Among the tested locations, mega-environments were identified with a set of hybrid combinations cultivated to gain the highest grain yield. The mega-environments of Gazipur (E2) and Ishwardi (E3) had a minimum discrimination ability and were most representative among all tested locations. BRRI hybrid dhan5, BRRI99A × BRRI38R, BRRI99A × BRRI45R, BRRI99A × BRRI31R, IR79156A × BRRI38R, BRRI99A × BRRI36R, BRRI99A × BRRI49R, IR79156A × BRRI31R showed the best performance in the mega-environments. The location of Barisal (E1) was favorable for IR79156A × BRRI45R, H-386, and SL8H. The hybrid combinations JhonokRaj and Heera-2 were moderately high yielders, and their position was near the origin, so they showed stale performance. The hybrid combination BRRI99A × BRRI38R was medium slender, and BRRI99A × BRRI36R and IR79156A × R line7 were long slender grain types, which are preferable to Bangladeshi consumers. These hybrids have the potential to be used for commercial cultivation in the locations. Stability analysis and genotype × environment interaction could be expanded even further for stress tolerance, stress resistance, grain quality, and nutritional content for precise identification of superior hybrid combinations.

## Figures and Tables

**Figure 1 plants-11-02336-f001:**
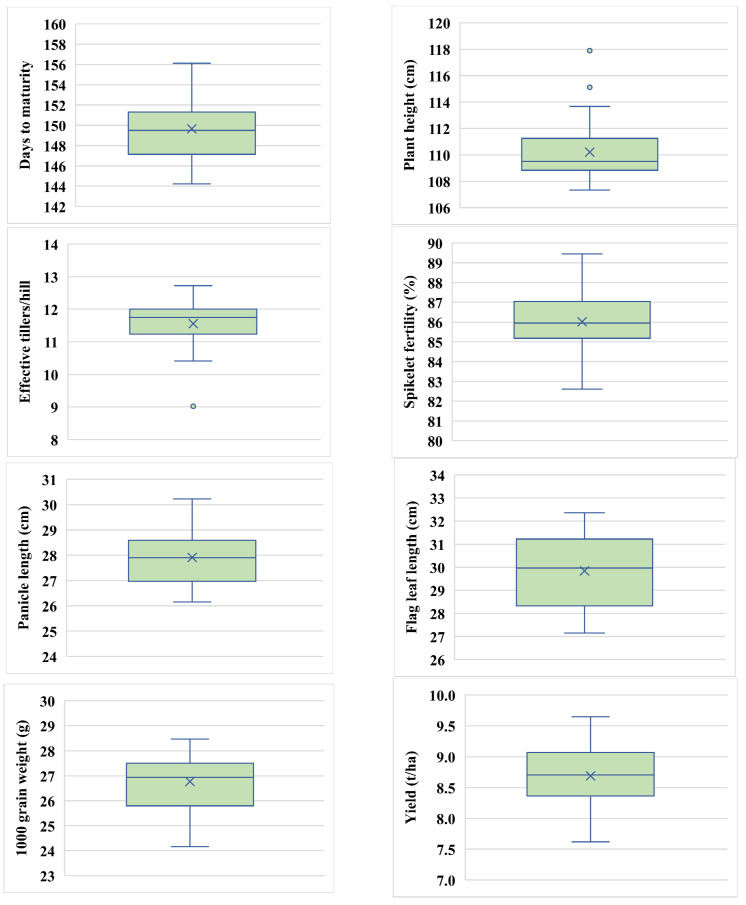
Boxplots showing descriptive statistics of 26 hybrid combinations.

**Figure 2 plants-11-02336-f002:**
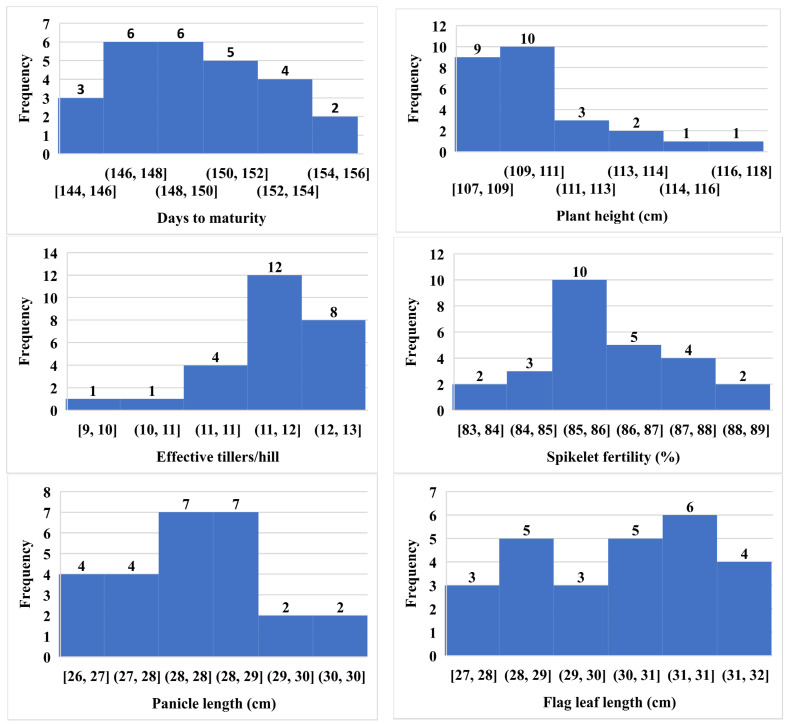
Histogram showing the frequency distribution of 26 hybrid combinations.

**Figure 3 plants-11-02336-f003:**
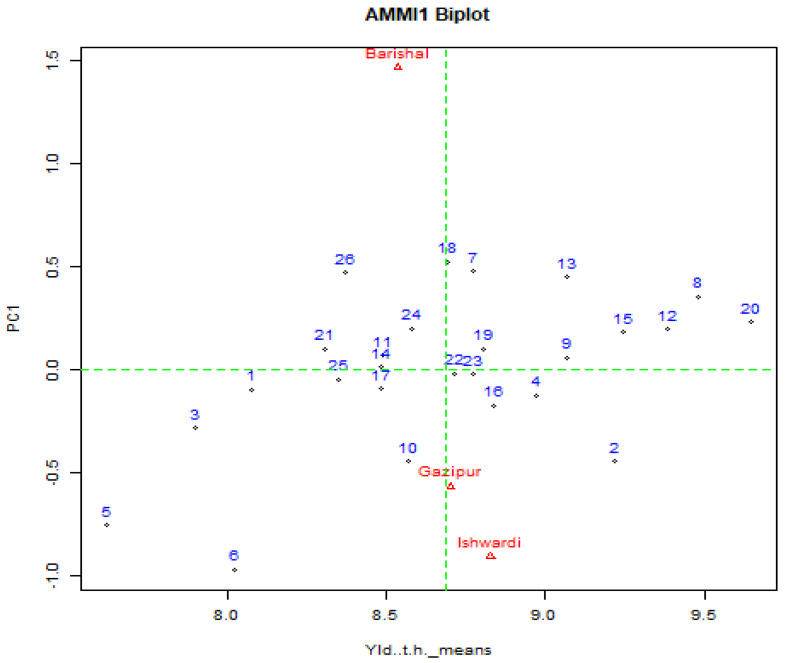
AMMI biplot (genotypes and environments plotted against their IPCA1 scores) for the grain yield of hybrid combinations.

**Figure 4 plants-11-02336-f004:**
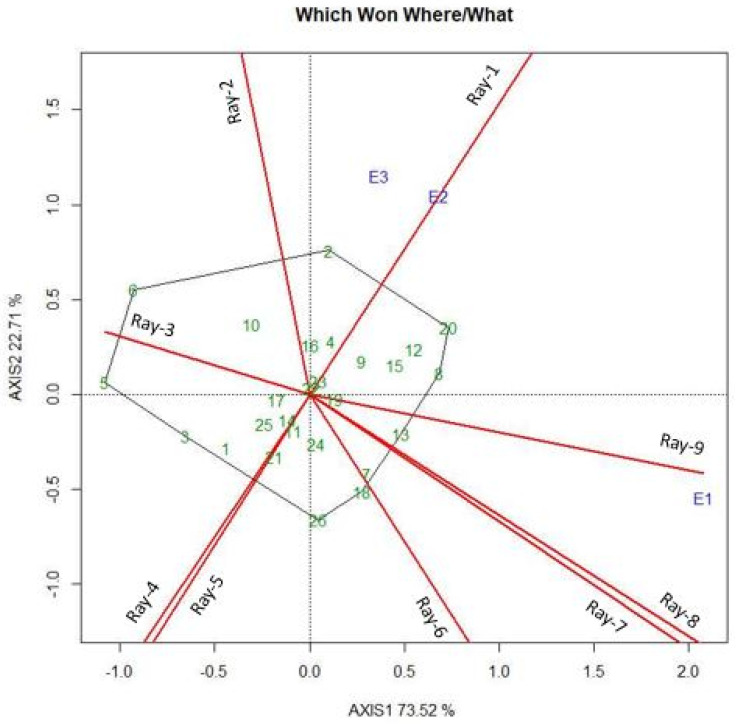
The polygon (mega-environment) view of the GGE biplot of 26 rice hybrid combinations was tested in three locations for grain yield.

**Figure 5 plants-11-02336-f005:**
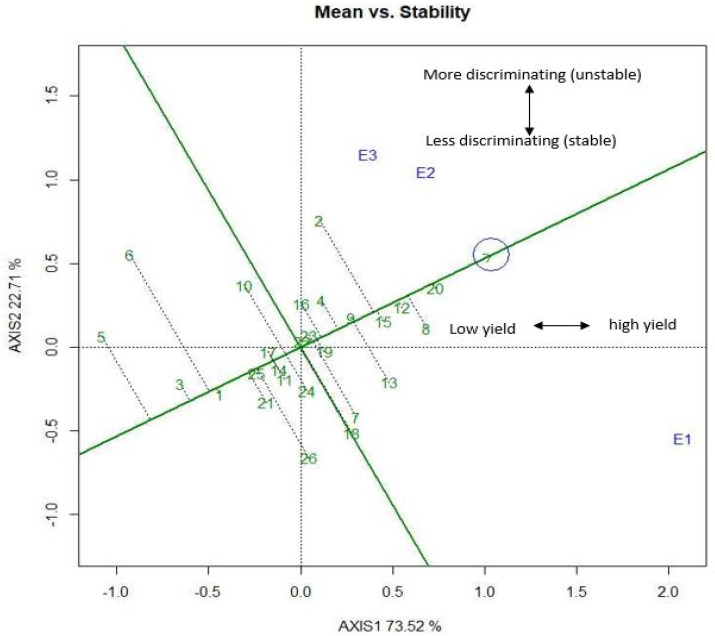
The vector view of the GGE biplot showing mean performance vs. stability of the hybrid combination.

**Figure 6 plants-11-02336-f006:**
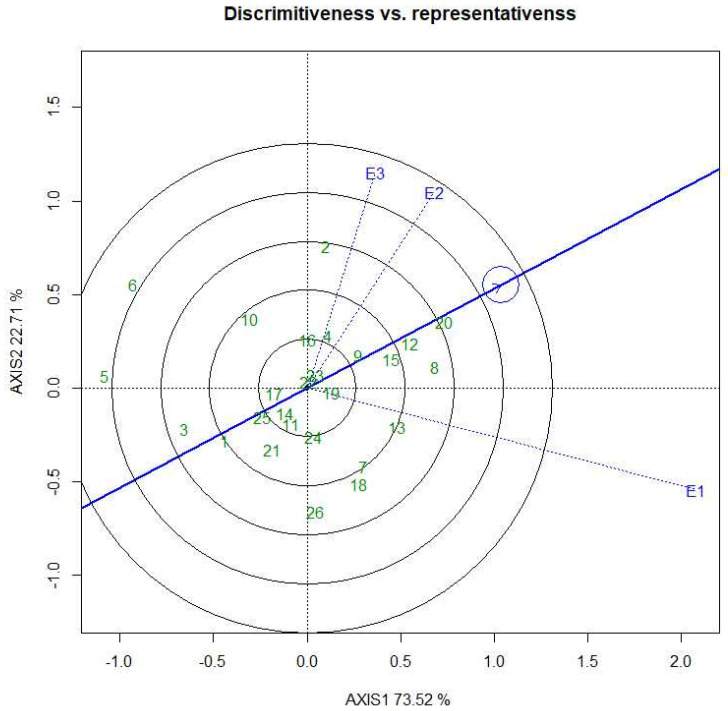
The vector view of the GGE biplot of 26 rice hybrid combinations in three locations for grain yield.

**Figure 7 plants-11-02336-f007:**
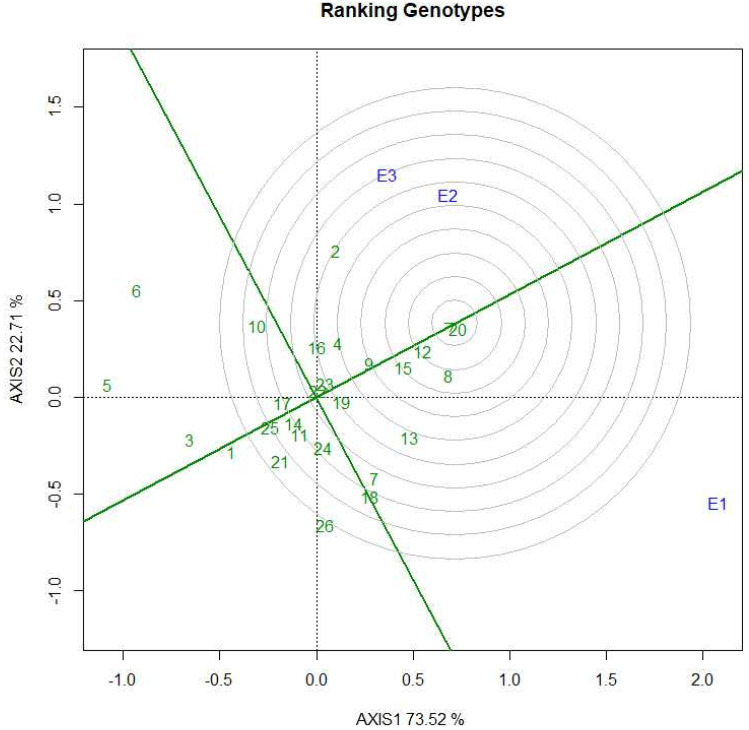
GGE biplot for ranking ideal rice hybrid combinations.

**Table 1 plants-11-02336-t001:** Descriptive statistics of yield and yield-related traits in some hybrid combinations.

Parameter	DTM	PH	ET/hill	SF%	PL	FLL	TGW	Grain Yield (t/ha)
Skewness	0.44	1.71	−1.69	0.09	0.25	−0.19	−0.41	−0.08
Kurtosis	−0.49	3.24	4.47	0.18	−0.17	−1.19	−0.18	−0.21

Legend: DTM = days to maturity; PH = plant height (cm); ET/hill = effective tillers/hills; SF (%) = spikelet fertility; PL (cm) = panicle length; FLL (cm) = flag leaf length; TGW (g) = thousand-grain weight.

**Table 2 plants-11-02336-t002:** Analysis of variance for grain yield and its related traits in some hybrid combination.

Source of Variance	df	DTM	PH (cm)	ET/ hill	SF (%)	PL	FLL	TGW	Grain Yield (t/ha)
Location	2	87.28 **	16.24 *	0.23 ^ns^	8.87 *	8.69 **	3.27 **	1.99 **	2.22 **
Variety	25	86.06 **	52.56 **	4.92 **	24.10 **	10.37 **	20.89 **	10.44 **	2.21 **
Location × Rep	6	2.91 ^ns^	4.03 ^ns^	1.59 *	12.54 **	1.06 **	2.41 **	0.24 *	0.30 **
Location × Variety	50	7.60 **	1.65 ^ns^	0.55 ^ns^	2.76 ^ns^	0.58 **	0.60 ^ns^	0.72 **	0.86 **
Error	150	2.73	3.66	0.45	2.19	0.22	0.55	0.12	0.08
Total	233								

^ns^ = Nonsignificant, ** Significant at 1% level, * significant at 5% level, DTM = days to maturity, PH = plant height (cm), ET/hill = effective tillers/hills, SF (%) = spikelet fertility percentage, FLL = flag leaf length (cm), TGW = thousand-grain weight (g), PL= panicle length (cm). Note: Plant height and effective tillers/hills were not normal. After logarithmic transformation they showed normal distribution.

**Table 3 plants-11-02336-t003:** Classification of hybrid combinations based on grain type.

Grain Type	Kernel Length (mm)	Kernel L × B Ratio	Hybrid Combination	Total
Medium slender	5.0–6.0	>3.0	BRRI48A × BRRI38R, BRRI99A × BRRI38R, IR79156A × BRRI38R, IR79156A × BRRI45R, TejGold, Teea, SL8H	7
Long slender	>6.0	>3.0	BRRI99A × BRRI36R, IR79156A × BRRI49R, IR79156A × R line7, BRRI99A × BRRI31R, IR79156A × BRRI31R, H-2264, H-386	7
Medium bold	5.0–6.0	2.0–3.0 or <3.0	BRRI35A × BRRI36R, BRRI35A × BRRI49R, BRRI99A × BRRI49R, BRRI35A × BRRI37R, BRRI35A × BRRI45R, BRRI99A × BRRI45R, BRRI35A × BRRI52R, BRRI hybrid dhan3, JhonokRaj, Heera-2, Gold	11
Long bold	>6.0	2.0–3.0 or <3.0	BRRI hybrid dhan5	1
		Total		26

**Table 4 plants-11-02336-t004:** Stability analysis for days to maturity of 26 hybrid combinations at three locations.

Sl No.	Combination	Environment				Pi	bi	S^2^di
		Barisal (E1)	Gazipur (E2)	Ishwardi (E3)	Overall Mean			
1	BRRI35A × BRRI36R	144.3	146.3	146.7	145.8	−3.3	0.631 *	0.00
2	BRRI99A × BRRI36R	144.0	148.7	149.3	147.3	−1.8	1.452	0.01
3	BRRI35A × BRRI49R	146.3	158.3	157.7	154.1	5	3.308	3.37
4	BRRI99A × BRRI49R	102.7	146.7	148.0	132.4	−16.7	12.802	18.61
5	IR79156A × BRRI49R	146.7	151.7	152.7	150.3	1.2	1.607	0.00
6	BRRI35A × BRRI37R	147.3	148.3	150.0	148.6	−0.5	0.584	0.91
7	BRRI48A × BRRI38R	149.3	150.0	149.3	149.6	0.5	0.072	0.26
8	BRRI99A × BRRI38R	150.0	150.7	150.3	150.3	1.2	0.131	0.09
9	IR79156A × BRRI38R	149.7	149.3	152.3	150.4	1.3	0.442	3.85
10	IR79156A × R line7	150.7	150.0	151.0	150.6	1.5	−0.011	0.52
11	BRRI35A × BRRI45R	152.7	155.0	157.3	155.0	5.9	1.084	1.49
12	BRRI99A × BRRI45R	153.3	153.3	155.7	154.1	5	0.418	2.24
13	IR79156A × BRRI45R	154.0	153.7	152.7	153.4	4.3	−0.274	0.37
14	BRRI35A × BRRI52R	147.3	147.0	147.3	147.2	−1.9	−0.035	0.07
15	BRRI99A × BRRI31R	147.3	150.3	149.0	148.9	−0.2	0.619	1.46
16	IR79156A × BRRI31R	145.3	147.7	147.0	146.7	−2.4	0.547	0.50
17	H-2264	148.7	149.3	150.7	149.6	0.5	0.429	0.61
18	H-386	149.7	152.0	151.0	150.9	1.8	0.488	0.84
19	BRRI hybrid dhan3	146.0	146.0	147.7	146.6	−2.5	0.298	1.15
20	BRRI hybrid dhan5	147.7	147.0	148.0	147.6	−1.5	−0.011	0.52
21	TejGold	145.3	145.7	147.0	146.0	−3.1	0.334	0.67
22	JhonokRaj	151.0	152.3	154.3	152.6	3.5	0.739	1.27
23	Heera-2	155.3	156.7	156.3	156.1	7	0.322	0.14
24	Gold	148.3	148.7	149.3	148.8	−0.3	0.215	0.16
25	Teea	144.0	144.0	144.7	144.2	−4.9	0.120	0.19
26	SL8H	150.0	150.0	148.3	149.4	0.3	−0.298	1.15
	Mean	149.9	146.8	150.5	149.1			
	Ij	0.8	−2.3	1.4				
	CV%	9.9	1.1	1.0				
	LSD (0.05)	23.7	2.76	2.50				

Pi, phenotypic index; *, bi, regression coefficient; S^2^di, deviation from regression; Ij, environmental index; *p* < 0.05.

**Table 5 plants-11-02336-t005:** Stability analysis for plant height of 26 hybrid combinations at three locations.

Sl No.	Combination	Environment				Pi	bi	S^2^di
		Barisal (E1)	Gazipur (E2)	Ishwardi (E3)	Overall Mean			
1	BRRI35A × BRRI36R	107.3	107.3	107.7	107.4	−2.8	0.377	0.01
2	BRRI99A × BRRI36R	105.7	107.0	109.3	107.3	−2.9	2.778	3.67
3	BRRI35A × BRRI49R	108.3	109.0	109.3	108.9	−1.3	0.439	0.44
4	BRRI99A × BRRI49R	108.3	107.3	109.7	108.4	−1.8	2.563 *	0.00
5	IR79156A × BRRI49R	109.7	110.3	113.3	111.1	0.9	3.476	2.60
6	BRRI35A × BRRI37R	108.7	107.3	108.3	108.1	−2.1	1.013	0.53
7	BRRI48A × BRRI38R	109.0	109.3	109.0	109.1	−1.1	−0.351	0.02
8	BRRI99A × BRRI38R	110.3	108.3	108.7	109.1	−1.1	0.193	2.28
9	IR79156A × BRRI38R	111.3	111.3	112.7	111.8	1.6	1.516	0.23
10	IR79156A × R line7	113.3	111.3	114.3	113.7	3.5	1.137	0.13
11	BRRI35A × BRRI45R	110.3	109.7	110.0	110.0	−0.2	0.316	0.18
12	BRRI99A × BRRI45R	108.7	18.3	110.3	109.1	−1.1	2.245	0.20
13	IR79156A × BRRI45R	111.0	112.3	111.7	111.7	1.5	−0.638	0.72
14	BRRI35A × BRRI52R	109.3	109.3	110.3	109.7	−0.5	1.137	0.13
15	BRRI99A × BRRI31R	109.0	107.3	109.7	108.7	−1.5	2.501	0.28
16	IR79156A × BRRI31R	112.7	112.7	113.0	112.8	2.6	0.377	0.01
17	H-2264	115.3	114.0	116.0	115.1	4.9	2.152	0.14
18	H-386	117.7	118.0	118.0	117.9	7.7	0.029	0.07
19	BRRI hybrid dhan3	110.0	109.7	110.0	109.9	−0.3	0.347	0.02
20	BRRI hybrid dhan5	110.0	109.3	110.3	109.9	−0.3	1.075	0.04
21	TejGold	110.0	108.3	108.7	109.0	−1.2	0.223	1.53
22	JhonokRaj	109.0	109.0	110.0	109.3	−0.9	1.137	0.13
23	Heera-2	111.0	109.7	109.7	110.1	−0.1	−0.125	1.18
24	Gold	110.7	109.3	109.7	109.9	−0.3	0.254	0.93
25	Teea	109.3	108.7	109.0	109.0	−1.2	0.316	0.18
26	SL8H	108.3	107.7	109.0	108.3	−1.9	1.455	0.01
	Mean	110.2	109.8	110.7	110.2			
	Ij	0	−0.4	0.5				
	CV%	1.8	1.9	1.5				
	LSD (0.05)	3.19	3.40	2.78				

Pi, phenotypic index; *, bi, regression coefficient; S^2^di, deviation from regression; Ij, environmental index; *p* < 0.05.

**Table 6 plants-11-02336-t006:** Stability analysis for effective tillers/hills of 26 hybrid combinations at three locations.

Sl No.	Combination	Environment				Pi	bi	S^2^di
		Barisal (E1)	Gazipur (E2)	Ishwardi (E3)	Overall Mean			
1	BRRI35A × BRRI36R	11.37	10.67	10.67	10.90	−0.65	7.418 *	0.00
2	BRRI99A × BRRI36R	11.77	11.63	10.90	11.43	−0.12	5.061	0.28
3	BRRI35A × BRRI49R	8.800	8.800	9.467	9.022	−2.528	−3.318	0.23
4	BRRI99A × BRRI49R	12.47	12.20	11.80	12.16	0.61	4.815	0.09
5	IR79156A × BRRI49R	11.00	11.10	11.77	11.29	−0.26	−4.377	0.23
6	BRRI35A × BRRI37R	9.833	10.70	10.70	10.41	−1.14	−9.186 *	0.00
7	BRRI48A × BRRI38R	12.17	11.87	11.30	11.78	0.23	5.998	0.17
8	BRRI99A × BRRI38R	10.63	10.80	10.73	10.72	−0.83	−1.436	0.00
9	IR79156A × BRRI38R	10.13	11.80	12.27	11.40	−0.15	−19.987	0.15
10	IR79156A × R line7	10.27	11.00	11.93	11.07	−0.48	−12.417	0.48
11	BRRI35A × BRRI45R	11.57	11.67	11.40	11.54	−0.01	0.266	0.04
12	BRRI99A × BRRI45R	12.10	11.93	11.83	11.96	0.41	2.263	0.01
13	IR79156A × BRRI45R	12.20	12.47	11.97	12.21	0.66	−0.340	0.12
14	BRRI35A × BRRI52R	12.23	11.97	11.77	11.99	0.44	3.820	0.02
15	BRRI99A × BRRI31R	12.50	11.87	11.17	11.84	0.29	10.194	0.27
16	IR79156A × BRRI31R	11.43	10.93	11.57	11.31	−0.24	2.148	0.20
17	H-2264	11.93	11.73	11.97	11.88	0.33	0.958	0.03
18	H-386	12.93	11.80	11.97	12.23	0.68	11.182	0.01
19	BRRI hybrid dhan3	12.17	12.27	11.87	12.10	0.55	0.929	0.08
20	BRRI hybrid dhan5	12.00	11.93	11.97	11.97	0.42	0.540	0.00
21	TejGold	12.80	12.77	12.60	12.72	1.17	1.182	0.01
22	JhonokRaj	12.07	12.00	12.07	12.04	0.49	0.374	0.00
23	Heera-2	12.00	11.47	11.60	11.69	0.14	4.988	0.01
24	Gold	12.33	11.77	11.87	11.99	0.44	5.508	0.00
25	Teea	11.33	10.87	10.87	11.02	−0.53	4.945 *	0.00
26	SL8H	12.00	11.53	11.63	11.72	0.17	4.448	0.00
	Mean	11.62	11.52	11.52	11.55			
	Ij	0.07	−0.03	−0.03				
	CV%	6.0	5.6	5.8				
	LSD (0.05)	1.14	1.07	1.09				

Pi, phenotypic index; *, bi, regression coefficient; S^2^di, deviation from regression; Ij, environmental index; *p* < 0.05.

**Table 7 plants-11-02336-t007:** Stability analysis for spikelet fertility percentage of 26 hybrid combinations at three locations.

Sl No.	Combination	Environment				Pi	bi	S^2^di
		Barisal (E1)	Gazipur (E2)	Ishwardi (E3)	Overall Mean			
1	BRRI35A × BRRI36R	86.17	85.60	85.10	85.62	−0.39	−1.559	0.02
2	BRRI99A × BRRI36R	86.67	89.63	90.97	89.09	3.08	6.152	1.08
3	BRRI35A × BRRI49R	85.30	84.93	85.60	85.28	−0.73	0.554	0.15
4	BRRI99A × BRRI49R	87.07	86.50	88.30	87.29	1.28	2.070	0.72
5	IR79156A × BRRI49R	80.73	83.50	85.17	83.13	−2.88	6.405	0.70
6	BRRI35A × BRRI37R	85.33	87.33	87.33	86.67	0.66	2.730	0.97
7	BRRI48A × BRRI38R	89.50	86.50	86.10	87.37	1.36	−4.720	1.84
8	BRRI99A × BRRI38R	85.13	85.43	87.50	86.02	0.01	3.673	0.25
9	IR79156A × BRRI38R	86.43	87.10	86.83	86.79	0.78	0.491	0.17
10	IR79156A × R line7	83.37	86.40	86.40	85.39	−0.62	4.139	2.24
11	BRRI35A × BRRI45R	85.73	84.73	84.73	85.07	−0.94	−1.361	0.25
12	BRRI99A × BRRI45R	87.27	87.77	87.77	87.60	1.59	0.684	0.06
13	IR79156A × BRRI45R	86.30	85.73	85.73	85.92	−0.09	−0.770	0.08
14	BRRI35A × BRRI52R	85.80	85.23	85.23	85.42	−0.59	−0.770	0.08
15	BRRI99A × BRRI31R	86.90	86.90	87.07	86.96	0.95	0.265	0.00
16	IR79156A × BRRI31R	85.80	85.40	85.40	85.53	−0.48	−0.543	0.04
17	H-2264	83.17	81.83	82.83	82.61	−3.4	−0.238	0.95
18	H-386	83.47	83.37	84.83	83.89	−2.12	2.180	0.26
19	BRRI hybrid dhan3	88.53	88.47	87.80	88.27	2.26	−1.141	0.03
20	BRRI hybrid dhan5	89.40	89.30	89.63	89.44	3.43	0.392	0.02
21	TejGold	85.60	86.60	86.70	86.30	0.29	1.524	0.21
22	JhonokRaj	85.53	86.53	87.50	86.52	0.51	2.891	0.03
23	Heera-2	84.67	86.60	86.67	85.98	−0.03	2.744	0.87
24	Gold	85.83	84.93	84.87	85.21	−0.8	−1.330	0.18
25	Teea	84.03	84.03	84.90	84.32	−1.69	1.370	0.07
26	SL8H	84.50	84.67	84.67	84.61	−1.4	0.230	0.01
	Mean	85.70	85.96	86.37	86.01			
	Ij	−0.31	−0.05	0.36				
	CV%	1.5	1.9	1.8				
	LSD (0.05)	2.09	2.65	2.50				

Pi, phenotypic index; bi, regression coefficient; S^2^di, deviation from regression; Ij, environmental index.

**Table 8 plants-11-02336-t008:** Stability analysis for panicle length of 26 hybrid combinations at three locations.

Sl No.	Combination	Environment				Pi	bi	S^2^di
		Barisal (E1)	Gazipur (E2)	Ishwardi (E3)	Overall Mean			
1	BRRI35A × BRRI36R	26.20	27.17	27.23	26.87	−1.04	1.707	0.02
2	BRRI99A × BRRI36R	27.43	29.17	30.43	29.01	1.1	4.414	0.19
3	BRRI35A × BRRI49R	25.43	27.80	27.80	27.01	−0.9	3.988	0.19
4	BRRI99A × BRRI49R	27.90	28.57	28.90	28.46	0.55	1.516	0.01
5	IR79156A × BRRI49R	25.50	26.23	26.73	26.16	−1.75	1.825	0.03
6	BRRI35A × BRRI37R	25.13	27.13	27.13	26.47	−1.44	3.370	0.13
7	BRRI48A × BRRI38R	28.77	28.50	28.40	28.56	0.65	−0.567 *	0.00
8	BRRI99A × BRRI38R	30.17	30.17	30.33	30.22	2.31	0.196	0.01
9	IR79156A × BRRI38R	27.60	27.60	27.90	27.70	−0.21	0.354	0.03
10	IR79156A × R line7	26.87	28.20	28.20	27.76	−0.15	2.247	0.06
11	BRRI35A × BRRI45R	26.70	27.03	27.03	26.92	−0.99	0.562	0.00
12	BRRI99A × BRRI45R	28.20	28.93	28.93	28.69	0.78	1.236	0.02
13	IR79156A × BRRI45R	28.77	29.07	29.07	28.97	1.06	0.505	0.00
14	BRRI35A × BRRI52R	26.80	27.07	27.07	26.98	−0.93	0.449	0.00
15	BRRI99A × BRRI31R	28.30	28.40	28.33	28.34	0.43	0.090	0.00
16	IR79156A × BRRI31R	27.73	27.77	27.77	27.76	−0.15	0.056	0.00
17	H-2264	27.73	27.80	27.87	27.80	−0.11	0.191 *	0.00
18	H-386	28.17	28.53	28.50	28.40	0.49	0.578	0.01
19	BRRI hybrid dhan3	28.07	28.13	28.47	28.22	0.31	0.505	0.03
20	BRRI hybrid dhan5	30.03	30.03	30.17	30.08	2.17	0.157	0.01
21	TejGold	26.07	26.07	26.33	26.16	−1.75	0.314	0.03
22	JhonokRaj	28.73	28.73	29.00	28.82	0.91	0.314	0.03
23	Heera-2	27.83	28.07	28.13	28.01	0.1	0.472	0.00
24	Gold	27.90	27.90	28.10	27.97	0.06	0.236	0.01
25	Teea	26.13	26.80	26.93	26.62	−1.29	1.281	0.00
26	SL8H	27.83	27.83	27.83	27.83	−0.08	0.000	0.00
	Mean	27.54	28.03	28.18	27.91			
	Ij	−0.37	0.12	0.27				
	CV%	1.5	1.8	1.8				
	LSD (0.05)	0.67	0.81	0.83				

Pi, phenotypic index; *, bi, regression coefficient; S^2^di, deviation from regression; Ij, environmental index; *p* < 0.05.

**Table 9 plants-11-02336-t009:** Stability analysis for flag leaf length of 26 hybrid combinations at three locations.

Sl No.	Combination	Environment				Pi	bi	S^2^di
		Barisal (E1)	Gazipur (E2)	Ishwardi (E3)	Overall Mean			
1	BRRI35A × BRRI36R	29.43	30.60	30.43	30.16	0.32	2.656	0.20
2	BRRI99A × BRRI36R	32.00	31.33	31.33	31.56	1.72	−1.732	0.04
3	BRRI35A × BRRI49R	27.00	27.23	27.23	27.16	−2.68	0.606	0.01
4	BRRI99A × BRRI49R	30.83	30.50	30.50	30.61	0.77	−0.866	0.01
5	IR79156A × BRRI49R	28.13	28.13	28.13	28.13	−1.71	0.000	0.00
6	BRRI35A × BRRI37R	26.37	29.40	29.40	28.39	−1.45	7.881	0.92
7	BRRI48A × BRRI38R	30.30	30.30	30.30	30.30	0.46	0.000	0.00
8	BRRI99A × BRRI38R	30.80	30.80	30.80	30.80	0.96	0.000	0.00
9	IR79156A × BRRI38R	31.23	31.23	31.23	31.23	1.39	0.000	0.00
10	IR79156A × R line7	29.23	29.23	29.23	29.23	−0.61	0.000	0.00
11	BRRI35A × BRRI45R	29.33	30.00	30.00	29.78	−0.06	1.732	0.04
12	BRRI99A × BRRI45R	31.27	31.60	31.60	31.49	1.65	0.866	0.01
13	IR79156A × BRRI45R	31.40	31.13	31.13	31.22	1.38	−0.693	0.01
14	BRRI35A × BRRI52R	27.37	28.43	28.43	28.08	−1.76	2.771	0.11
15	BRRI99A × BRRI31R	32.37	32.33	32.37	32.36	2.52	−0.012	0.00
16	IR79156A × BRRI31R	31.43	31.43	31.60	31.49	1.65	0.374	0.01
17	H-2264	31.93	32.40	31.37	31.90	2.06	−1.109	0.43
18	H-386	29.47	29.67	30.13	29.76	−0.08	1.568	0.03
19	BRRI hybrid dhan3	28.53	28.60	30.03	29.06	−0.78	3.393	0.47
20	BRRI hybrid dhan5	31.07	31.07	31.33	31.16	1.32	0.599	0.02
21	TejGold	27.83	27.83	28.17	27.94	−1.9	0.749	0.03
22	JhonokRaj	27.77	27.77	28.67	28.07	−1.77	2.022	0.20
23	Heera-2	28.47	28.80	28.87	28.71	−1.13	1.015	0.01
24	Gold	30.50	30.50	31.03	30.68	0.84	1.198	0.07
25	Teea	29.10	29.27	29.67	29.34	−0.5	1.331	0.02
26	SL8H	27.07	27.07	27.80	27.31	−2.53	1.647	0.13
	Mean	29.62	29.87	30.03	29.84			
	Ij	−0.22	0.03	0.19				
	CV%	2.6	2.5	2.3				
	LSD (0.05)	1.26	1.24	1.14				

Pi, phenotypic index; bi, regression coefficient; S^2^di, deviation from regression; Ij, environmental index.

**Table 10 plants-11-02336-t010:** Stability analysis for the thousand-grain weight of 26 hybrid combinations at three locations.

Sl No.	Combination	Environment				Pi	bi	S^2^di
		Barisal (E1)	Gazipur (E2)	Ishwardi (E3)	Overall Mean			
1	BRRI35A × BRRI36R	27.10	26.73	26.57	26.80	0.03	−1.706	0.00
2	BRRI99A × BRRI36R	28.03	28.30	28.57	28.30	1.53	1.611	0.01
3	BRRI35A × BRRI49R	25.47	28.13	28.13	27.24	0.47	9.297	0.32
4	BRRI99A × BRRI49R	26.47	26.47	27.97	26.97	0.2	3.837	0.75
5	IR79156A × BRRI49R	23.40	24.30	24.80	24.17	−2.6	4.416	0.01
6	BRRI35A × BRRI37R	24.60	26.10	26.17	25.62	−1.15	5.400	0.08
7	BRRI48A × BRRI38R	25.57	25.57	25.57	25.57	−1.2	−0.001	0.00
8	BRRI99A × BRRI38R	26.63	27.47	27.47	27.19	0.42	2.904	0.03
9	IR79156A × BRRI38R	26.70	27.23	27.23	27.06	0.29	1.858	0.01
10	IR79156A × R line7	25.30	25.30	25.53	25.38	−1.39	0.596	0.02
11	BRRI35A × BRRI45R	25.70	25.93	25.93	25.86	−0.91	0.812	0.00
12	BRRI99A × BRRI45R	27.77	27.83	27.83	27.81	1.04	0.231 *	0.00
13	IR79156A × BRRI45R	28.77	26.40	26.40	27.02	0.25	−6.510	0.16
14	BRRI35A × BRRI52R	26.33	26.53	26.17	26.34	−0.43	−0.242	0.06
15	BRRI99A × BRRI31R	28.40	28.50	28.50	28.47	1.7	0.348	0.00
16	IR79156A × BRRI31R	26.37	26.37	26.37	26.37	−0.4	−0.001	0.00
17	H-2264	27.37	26.67	26.67	26.90	0.13	−2.442	0.02
18	H-386	26.50	26.33	26.33	26.39	−0.38	−0.582	0.00
19	BRRI hybrid dhan3	27.27	27.47	27.47	27.40	0.63	0.696	0.00
20	BRRI hybrid dhan5	28.27	28.27	28.13	28.22	1.45	−0.342	0.01
21	TejGold	25.47	25.47	25.47	25.47	−1.3	−0.001	0.00
22	JhonokRaj	27.50	28.50	28.30	28.10	1.33	2.974	0.11
23	Heera-2	27.80	28.00	28.10	27.97	1.2	0.952	0.00
24	Gold	26.93	27.20	27.30	27.14	0.37	1.184	0.00
25	Teea	26.83	26.83	26.73	26.80	0.03	−0.257	0.00
26	SL8H	25.33	25.33	25.70	25.46	−1.31	0.937	0.04
	Mean	26.59	26.82	26.90	26.77			
	Ij	−0.18	0.05	0.13				
	CV%	1.1	1.3	1.5				
	LSD (0.05)	0.49	0.56	0.66				

Pi, phenotypic index; *, bi, regression coefficient; S^2^di, deviation from regression; Ij, environmental index; *p* < 0.05.

**Table 11 plants-11-02336-t011:** Stability analysis for grain yield of 26 hybrid combinations at three locations.

Sl No.	Combination	Environment				Pi	bi	S^2^di
		Barisal (E1)	Gazipur (E2)	Ishwardi (E3)	Overall Mean			
1	BRRI35A × BRRI36R	7.743	8.173	8.310	8.076	−0.613	1.714	0.01
2	BRRI99A × BRRI36R	8.300	9.673	9.673	9.216	0.527	4.244	0.23
3	BRRI35A × BRRI49R	7.313	7.757	8.623	7.898	−0.791	3.814	0.06
4	BRRI99A × BRRI49R	8.640	8.790	9.490	8.973	0.284	2.438	0.07
5	IR79156A × BRRI49R	6.297	7.630	8.930	7.619	−1.07	7.787	0.02
6	BRRI35A × BRRI37R	6.253	8.853	8.963	8.023	−0.666	8.345	0.74
7	BRRI48A × BRRI38R	9.370	8.487	8.460	8.772	0.083	−2.805	0.09
8	BRRI99A × BRRI38R	9.927	8.927	9.583	9.479	0.79	−1.238	0.43
9	IR79156A × BRRI38R	8.977	9.143	9.087	9.069	0.38	0.355	0.01
10	IR79156A × R line7	7.650	9.063	8.997	8.570	−0.119	4.179	0.28
11	BRRI35A × BRRI45R	8.420	8.520	8.523	8.488	−0.201	0.319	0.00
12	BRRI99A × BRRI45R	9.510	9.417	9.233	9.387	0.698	−0.805	0.00
13	IR79156A × BRRI45R	9.623	8.817	8.770	9.070	0.381	−2.624	0.07
14	BRRI35A × BRRI52R	8.317	8.590	8.543	8.483	−0.206	0.713	0.01
15	BRRI99A × BRRI31R	9.367	9.103	9.257	9.242	0.553	−0.381	0.03
16	IR79156A × BRRI31R	8.380	8.980	9.150	8.837	0.148	2.334	0.02
17	H-2264	8.143	8.687	8.617	8.482	−0.207	1.482	0.05
18	H-386	9.363	8.383	8.333	8.693	0.004	−3.169	0.10
19	BRRI hybrid dhan3	8.793	8.780	8.847	8.807	0.118	0.147	0.00
20	BRRI hybrid dhan5	9.857	9.470	9.617	9.648	0.959	−0.781	0.04
21	TejGold	8.287	8.320	8.313	8.307	−0.382	0.084 *	0.00
22	JhonokRaj	8.500	8.810	8.833	8.714	0.025	1.024	0.01
23	Heera-2	8.557	8.873	8.890	8.773	0.084	1.026	0.01
24	Gold	8.713	8.553	8.467	8.578	−0.111	−0.739 *	0.00
25	Teea	8.087	8.457	8.497	8.347	−0.342	1.256	0.01
26	SL8H	8.960	8.060	8.083	8.368	−0.321	−2.715	0.11
	Mean	8.513	8.704	8.850	8.689			
	Ij	−0.176	0.015	0.161	0			
	CV%	2.7	3.2	3.7				
	LSD (0.05)	0.37	0.45	0.53				

Pi, phenotypic index; *, bi, regression coefficient; S^2^di, deviation from regression; Ij, environmental index; *p* < 0.05.

**Table 12 plants-11-02336-t012:** List of 26 rice hybrid combinations.

Sl No.	Combination	Sl No.	Combination	Sl No.	Combination
1	BRRI35A × BRRI36R	10	IR79156A × R line7	19	BRRI hybrid dhan3
2	BRRI99A × BRRI36R	11	BRRI35A × BRRI45R	20	BRRI hybrid dhan5
3	BRRI35A × BRRI49R	12	BRRI99A × BRRI45R	21	TejGold
4	BRRI99A × BRRI49R	13	IR79156A × BRRI45R	22	JhonokRaj
5	IR79156A × BRRI49R	14	BRRI35A × BRRI52R	23	Heera-2
6	BRRI35A × BRRI37R	15	BRRI99A × BRRI31R	24	Gold
7	BRRI48A × BRRI38R	16	IR79156A × BRRI31R	25	Teea
8	BRRI99A × BRRI38R	17	H-2264	26	SL8H
9	IR79156A × BRRI38R	18	H-386		

## Data Availability

Data recorded in the current study are available in all tables and figures of the manuscript.

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
