# Peer review of "Assessment of GGE, AMMI, Regression, and Its Deviation Model to Identify Stable Rice Hybrids in Bangladesh"

_plants, 2022, doi:10.3390/plants11182336_

Round 1

Reviewer 1 Report (Previous Reviewer 1)

Some sentences are not easy to understand, so please proofread whole manuscript carefully to remove linguistic mistakes, such as in Abstract

L29-31: Please re-phrase

L32-35 better to mention as “Among the tested locations, Gazipur (E2) and Ishwardi (E3) were identified as mega environments for the hybrid combinations, including BRRI99A × BRRI36R, BRRI99A × BRRI49R, IR79156A × BRRI31R, IR79156A × BRRI38R, BRRI hybrid dhan5, BRRI99A × BRRI38R, BRRI99A × BRRI45R, and BRRI99A × BRRI31R based on their average action and fixity.

L41: add location names

Figures are poor quality, so need to improve, such as Fig. 1: too big, please compress it on a single page. Remove top legend, such as Day to maturity, yield, 1000-grain weight. These legends are present along Y-axis, so just a repetition and remove from top

L358-Re-phrase

Author Response

Reviewer 1 (Round 1)

Comment: Some sentences are not easy to understand, so please proofread whole manuscript carefully to remove linguistic mistakes, such as in Abstract

Author response: We would like to appreciate and thank honorable Reviewer 1 for giving valuable time and critically reviewing our MS for its substantial improvement and also, for your kind appreciation and positive decision. We checked our whole manuscript by the English expert team of our university

Comment: L29-31: Please re-phrase

Author response: We would like to appreciate and thank honorable Reviewer 1. We have rephrased it.

Comment: L32-35 better to mention as “Among the tested locations, Gazipur (E2) and Ishwardi (E3) were identified as mega environments for the hybrid combinations, including BRRI99A × BRRI36R, BRRI99A × BRRI49R, IR79156A × BRRI31R, IR79156A × BRRI38R, BRRI hybrid dhan5, BRRI99A × BRRI38R, BRRI99A × BRRI45R, and BRRI99A × BRRI31R based on their average action and fixity.

Author response: We would like to appreciate and thank honorable Reviewer 1. We have changed it according to your suggestion.

Comment: L41: add location names

Author response: We would like to appreciate and thank honorable Reviewer 1. We have added the location name.

Comment: Figures are poor quality, so need to improve, such as Fig. 1: too big, please compress it on a single page. Remove top legend, such as Day to maturity, yield, 1000-grain weight. These legends are present along Y-axis, so just a repetition and remove from top

Author response: We would like to appreciate and thank honorable Reviewer 1. We have redrawn Figure 1 and accommodated it in a single page. The top legend, such as Day to maturity, yield, 1000-grain weight, etc. were removed from figure 1.

Comment: L358-Re-phrase

 Author response: We would like to appreciate and thank honorable Reviewer 1. We have rephrased it.

Reviewer 2 Report (Previous Reviewer 3)

I have go through the revised manuscript. the manuscript has

been significantly improved and could be accepted for publication.

Author Response

Reviewer 2 (Round 1)

Comment: I have go through the revised manuscript. the manuscript has been significantly improved and could be accepted for publication.

Author response: We would like to appreciate and thank honorable Reviewer 2 for giving valuable time and critically reviewing our MS for its substantial improvement and also, for your kind appreciation and positive decision to publish.

This manuscript is a resubmission of an earlier submission. The following is a list of the peer review reports and author responses from that submission.

Round 1

Reviewer 1 Report

The manuscript entitled “Identification of Stable Rice Hybrids in Different Locations of 2 Bangladesh” is focused on the performance of different hybrids under various environments. The results are interesting and useful for rice breeders. However, I would like to suggest few changes in the manuscript.

There is no news about the ecological zones, where these hybrids were planted. Authors should provide climatic conditions of these zones.

1.      Please rewrite the sentence in L39

2.      Add a reference in first paragraph of introduction

3.      L196,199, it’s not good to say results were identical or support this. Please compare your results with those studies, so readers can understand in a better way.

4.      L319: tall cruel efficiency?

Author Response

Reviewer 1 (Round 1)

Comment: The manuscript entitled “Identification of Stable Rice Hybrids in Different Locations of 2 Bangladesh” is focused on the performance of different hybrids under various environments. The results are interesting and useful for rice breeders. However, I would like to suggest few changes in the manuscript.

Author response: We would like to appreciate and thank honorable Reviewer 1 for giving valuable time and critically reviewing our MS for its substantial improvement and your kind appreciation.

Comment: There is no news about the ecological zones, where these hybrids were planted. Authors should provide climatic conditions of these zones.

Author response: We would like to appreciate and thank honorable Reviewer 1. We have provided climatic conditions of these three zones according to your suggestion in supplementary Table S1.

Comment: 1. Please rewrite the sentence in L39

Author response: We would like to appreciate and thank honorable Reviewer 1. We have rewritten the sentence according to your suggestion.

Comment: 2. Add a reference in first paragraph of introduction

Author response: We would like to appreciate and thank honorable Reviewer 1. According to your suggestion, we have added a reference in the first paragraph of the introduction.

Comment: 3. L196,199, it’s not good to say results were identical or support this. Please compare your results with those studies, so readers can understand in a better way.

Author response: We would like to appreciate and thank honorable Reviewer 1. We have rephased the sentences for a better understanding of readers.

Comment: 4. L319: tall cruel efficiency?

Author response: We would like to appreciate and thank honorable Reviewer 1. Tall cruel efficiency means strong support of Which-Won-Where pattern for selection of environment of a hybrid rice variety

Reviewer 2 Report

The current manuscript presents a study on several rice hybrids in three locations of Bangladesh. The manuscript has a complex redaction and its read is very difficult for one non-specialist reader.

This study has an important statistical analysis, although its implication in the rice breeding or hybrids election results very complex. I think that the general interest for the readers of this journal could be very limited.

The title is scarcely identified of the true content of the manuscript and it should be changed.

Some part of the manuscript could be deleted. For example, the section 3.2 that is irrelevant for the information showed in this study. The descriptive data have scarce utility.

Some data have been unnecessary repeated. The table 2 and figure 2 shows the same data.

The measured traits should be included in the section 2.4.

Author Response

Reviewer 2 (Round 1)

Comment: The current manuscript presents a study on several rice hybrids in three locations of Bangladesh. The manuscript has a complex redaction and its read is very difficult for one non-specialist reader.

 Author response: We would like to appreciate and thank honorable Reviewer 2 for giving valuable time and critically reviewing our MS for its substantial improvement. For your kind consideration, in the case of consistency/stability (G × E interactions) analysis, we used most of the methods/terminologies that are related to strong genetic background. Without genetic background, it is difficult to understand clearly. However, according to your suggestions, we tried our best to improve our manuscript in the simplest way for the high readability of the general readers.

Comment: This study has an important statistical analysis, although its implication in the rice breeding or hybrids election results very complex. I think that the general interest for the readers of this journal could be very limited.

Author response: For your information and kind consideration, we submitted the manuscript in the Special Issue (SI): "Advances in Genetics and Breeding of Grain Crops” in the Journal: Plants. We evaluate the hybrids by different widely used, simple, and popular stability parameters/methods, including descriptive statistics, frequency distribution-based histogram, AMMI, and GGE biplots, Which-Won-Where pattern, etc. for identification of consistent/stable hybrids to cultivate all over Bangladesh or particular location by considering G × E interactions of hybrids in different location of Bangladesh. There are thousands of published articles for stability analysis using these methods. Nobody can’t study stability analysis without these methods. As an experienced breeder, I think the manuscript 100% matched this issue of the “plants” journal and 100% fulfilled the general interest of the agriculturist/plant breeders/agronomist of this journal. The editorial office and honorable guest editor were also found suitable for this SI of the journal. For this reason, they send the manuscript for peer review. The editorial office selects the appropriate/suitable manuscript and sends it for peer review, otherwise, they declined for publication before sending it to peer review.

Comment: The title is scarcely identified of the true content of the manuscript and it should be changed.

Author response: We would like to appreciate and thank honorable Reviewer 2. We have changed the title according to your suggestion.

Comment: Some part of the manuscript could be deleted. For example, the section 3.2 that is irrelevant for the information showed in this study. The descriptive data have scarce utility.

Author response: We would like to appreciate and thank honorable Reviewer 2. For your information and kind consideration, stability referrers to the consistency of a genotype(s) over a wide range of environments. Stability is the result of gene actions i. e., genotypes (s) and their interaction with environments (Genotype and environment interactions). Stability is directly related to gene actions of traits. So, skewness and kurtosis analysis are directly helpful to identify gene actions of a trait and find out the stable genotypes of a crop. As the other two reviewers don’t suggest to delete this subsection, so we retained it. However, for better clarification and understanding we have rephrased the sentences in subsection 3.2.

For instance, we are stating some information from published literature below:

Positive skewness is associated with complementary gene actions, while negative skewness is associated with duplicate (additive × additive) gene actions [Kamdar et al. (2020) Crop Breed. Appl. Biotechnol. 20 (2)].

Skewness is a measurement of symmetry of the population and is often used to indicate the nature of gene action for any trait (Fisher, 1932). Positive skewness corresponds to complementary gene action, whereas negative skewness corresponds to duplicate epistasis (Pooni et al., 1977). Kurtosis designates the number of genes that is governing the trait of interest (Robson,1956). Kurtosis is a measure of peakedness or tailness of a data and measuring whether it is right-tailed or left-tailed to the normal distribution. Kurtosis is manifested in three levels viz., leptokurtic (positive kurtosis) signifying that the traits are controlled by fewer genes, mesokurtic (normal distribution at 0), and platykurtic (negative kurtosis) signifying that the traits are controlled by a larger number of genes (Kapur, 1981).

The nature of gene action and the number of genes controlling the trait is usually measured by the critical analysis of distribution properties by third-order statistics such as skewness and kurtosis which are more important than the first and second-order statistics that unravel only the interaction effects (Rani et al., 2016). Skewness indicates the cluster of deviation above and below the value of central tendency and defines the extent of deviation in the distribution. Positive skewness would indicate the traits to be controlled by dominant and complementary gene action whereas a negative skewness would indicate the traits to be controlled by dominant and duplicate epistasis. Kurtosis indicates the level of peakedness over the population with a leptokurtic distribution would mean that the trait in question is controlled by fewer genes whereas a platykurtic distribution would mean that the trait is governed by many genes

Comment: Some data have been unnecessary repeated. The table 2 and figure 2 shows the same data.

Author response: We would like to appreciate and thank honorable Reviewer 2. We have deleted repeated data of  Figure 1 from Table 2 retaining the data of skewness and kurtosis only. 

Comment: The measured traits should be included in the section 2.4.

Author response: We would like to appreciate and thank honorable Reviewer 2. We have included measured traits in subsection 2.4.

Reviewer 3 Report

In this manuscript, the authors evaluated twenty-six rice hybrids at three locations. Agronomic traits were analyzed to dissect variety effects, location effects and variety* locations interactions. However, the manuscript is generally descriptive, the take-home message is lacking. This is a preliminary study.

For three locations, their latitude and longitude should be provided, and the weather data for these locations should also provided to assess environmental effects.

This manuscript is not well written with lots of grammatical mistakes in the manuscript, please try to improve the readability of your manuscript.

All abbreviations must be accompanied with the full name for their first presence in the manuscript, for example, RCBD, GEI,

In Table 1, from genotype 1 to 16, the parents for the crossing were provided, while from genotype 17 to 26, the parents for the hybrid crossing are missing.

Author Response

Reviewer 3 (Round 1)

Comment: In this manuscript, the authors evaluated twenty-six rice hybrids at three locations. Agronomic traits were analyzed to dissect variety effects, location effects and variety* locations interactions. However, the manuscript is generally descriptive, the take-home message is lacking. This is a preliminary study.

Author response: We would like to appreciate and thank honorable Reviewer 3 for giving valuable time and critically reviewing our MS for its substantial improvement. For your kind information and consideration, to identify stable genotypes, the GGE and AMMI model of analysis are modern (the old one is Eberhart and Russel’s model, 1966), well-accepted, popular, and widely used methods. There are a lot of published articles in the literature where these analyses were used.

Comment: For three locations, their latitude and longitude should be provided, and the weather data for these locations should also provided to assess environmental effects.

Author response: We would like to appreciate and thank honorable Reviewer 3. We have provided the latitude and longitude of three locations and weather data of these locations.

Comment: This manuscript is not well written with lots of grammatical mistakes in the manuscript, please try to improve the readability of your manuscript.

Author response: We would like to appreciate and thank honorable Reviewer 3. We have checked our manuscript with an English expert. Now I think, the readability of our manuscript was improved.  For your kind consideration, all MDPI journals finally edit the language of the accepted manuscript before publication.

Comment: All abbreviations must be accompanied with the full name for their first presence in the manuscript, for example, RCBD, GEI,

Author response: We would like to appreciate and thank honorable Reviewer 3. We have defined the full name of all abbreviations for the first appearance.

Comment: In Table 1, from genotype 1 to 16, the parents for the crossing were provided, while from genotype 17 to 26, the parents for the hybrid crossing are missing.

Author response: We would like to appreciate and thank honorable Reviewer 3. Genotypes 1 to 16 are the newly developed hybrid combinations waiting for release for commercial cultivation. Genotypes 17 to 26 were already released for commercial cultivation. These were used for the comparison of yield, stability, and other characteristics as a check variety.

Round 2

Reviewer 2 Report

In science, each author's background clearly influences any study he or she conducts. In my opinion, based on my previous comments, some things in this study should have been done differently. But of course this is my opinion, which is also influenced by my own background.

In any case, I think the authors have addressed most of the problems and reviewed the MS satisfactorily. I positively value the effort made by the authors for the revision of this manuscript.

Author Response

Comment: In science, each author's background clearly influences any study he or she conducts. In my opinion, based on my previous comments, some things in this study should have been done differently. But of course, this is my opinion, which is also influenced by my own background.

Author response: We would like to appreciate and thank honorable Reviewer 2 for giving valuable time and critically reviewing our MS for its substantial improvement and your kind appreciation.

Comment: In any case, I think the authors have addressed most of the problems and reviewed the MS satisfactorily. I positively value the effort made by the authors for the revision of this manuscript.

Author response: We would like to appreciate and thank honorable Reviewer 2 for accepting our manuscript for publication.

Reviewer 3 Report

I go through the revised manuscript. The authors carefully revised the manuscript. For the novelty, the authors answered that  they used the GGE and AMMI model  for analysis which are modern well-accepted, popular, and widely used methods. However, the novelty is based on the complex analysing method, but based on the inspiration brought to others based on their work. For example, for the abstract, the conclusion is descriptive.

Author Response

Comment: I go through the revised manuscript. The authors carefully revised the manuscript. For the novelty, the authors answered that they used the GGE and AMMI model for analysis which are modern well-accepted, popular, and widely used methods. However, the novelty is based on the complex analysing method, but based on the inspiration brought to others based on their work. For example, for the abstract, the conclusion is descriptive.

Author response: We would like to appreciate and thank honorable Reviewer 3 for giving valuable time and critically reviewing our MS for its substantial improvement and your kind appreciation.

We have rewritten the conclusive remarks descriptive and very simple way as per our findings with general future trend.
